# Gas Separation Membrane Module Modeling: A Comprehensive Review

**DOI:** 10.3390/membranes13070639

**Published:** 2023-06-30

**Authors:** Marcos Da Conceicao, Leo Nemetz, Joanna Rivero, Katherine Hornbostel, Glenn Lipscomb

**Affiliations:** 1Department of Chemical Engineering, University of Toledo, Toledo, OH 43606, USA; 2Department of Mechanical Engineering and Materials Science, University of Pittsburgh, Pittsburgh, PA 15213, USA

**Keywords:** membrane modeling, membrane gas separation, membrane module design

## Abstract

Membrane gas separation processes have been developed for diverse gas separation applications that include nitrogen production from air and CO_2_ capture from point sources. Membrane process design requires the development of stable and robust mathematical models that can accurately quantify the performance of the membrane modules used in the process. The literature related to modeling membrane gas separation modules and model use in membrane gas separation process simulators is reviewed in this paper. A membrane-module-modeling checklist is proposed to guide modeling efforts for the research and development of new gas separation membranes.

## 1. Introduction

This review summarizes the literature related to the development and use of membrane gas separation module models in process simulation. The manuscript is divided into the following sections:Introduction to membrane gas separations;Membrane module designs;Module flow patterns;Membrane modeling review;Modeling non-idealities in membrane gas separation modules;Gas separation process modeling applications and challenges.

A final section provides a modeling checklist for developing membrane module models that incorporate the physics and module design detail required to accurately predict performance. The overall goal is to provide the reader with the broad background needed to assess the use of membrane module models in gas separation process simulations.

## 2. Introduction to Membrane Gas Separations

Separation processes are a critical part of chemical and purified product production. Distillation is the main separation technology in industrial plants and is responsible for 10–15% of the world’s energy usage [1]. The energy-intensive nature of distillation has led both industry and academia to seek more efficient alternatives that can limit the rise of atmospheric carbon dioxide (CO_2_) concentration while still meeting the desired separation targets. The use of non-thermal separation technologies such as membranes has attracted interest in several industrial sectors due to their unique advantages, such as the following [2,3,4]:Simple operation and installation;No chemical usage;Low energy consumption;Facile scale-up due to process compactness and modularity;Flexible integration with other process units to form hybrid processes with reduced energy consumption.

Membrane-based gas separation has become a relevant process unit that can compete with conventional processes, such as distillation, absorption, and adsorption [4]. Binary and multicomponent gas separations are achievable with membranes, making them an ideal candidate for myriad applications including but not limited to nitrogen production from air, natural gas sweetening, hydrogen recovery, biogas upgrading, gas dehydration/drying, volatile organic compound (VOC) recovery, and CO_2_ capture from flue gas streams [5,6].

Membrane materials can be made from polymeric materials, carbon, inorganic metals and ceramics, or a combination (mixed matrix or hybrid materials) [5,7]. Polymeric membranes are used predominantly in industrial gas separations because they are inexpensive, easy to manufacture, and robust [5]. Polymeric membrane materials are subdivided into rubbery and glassy polymers. Glassy polymers typically possess lower permeabilities and higher selectivities than rubbery polymers due to reduced molecular motion that restricts fluctuations in free volume. This increases selectivity through enhanced molecular sieving, but it decreases gas diffusion coefficients [8,9].

For membranes that do not possess permanent porosity (i.e., a dense polymeric material), the most widely accepted model for transport is the solution–diffusion model. Transport is envisioned as occurring in three steps: (1) gas dissolution or sorption into the membrane on the high-pressure side of the material, (2) sorbed gas diffusion through the membrane, and (3) desorption from the membrane on the low-pressure side [10]. The driving force for transport is controlled by the chemical potential difference between the high and low pressure contacting as phases. The driving force is created by gas compression or vacuum. For ideal gas contacting phases, gas flux across the membrane is given by the following calculation:(1)$$J_{i}=\frac{Q_{i}}{\delta }\times P_{r}\times xi-Pp\times yi$$
where $J_{i}$ is the flux across the membrane of species i (mol/m^2^/s); $Q_{i}$ is the membrane permeability for species i (mol·m/m^2^/s/Pa); δ is the effective membrane thickness (m); $P_{r}$ and $P_{p}$ are the feed/retentate (high pressure) and permeate (low pressure) pressures (Pa), respectively; and $x_{i}$ and $y_{i}$ are the high- and low-pressure gas phase mole fractions (mol/mol), respectively. The permeability is equal to the product of gas solubility and diffusivity in the membrane. The ratio of permeability to membrane thickness in Equation (1) is defined as the gas permeance: $q\equiv \frac{Q_{i}}{\delta }$ (mol/m^2^/s/Pa). Commonly, permeance is expressed in gas permeation units (GPU) where 1 GPU = 3.35 × 10^−10^ (mol/m^2^/s/Pa).

High-performance polymeric membranes typically consist of a thick, dense submicron layer on top of a porous support, as shown in Figure 1. Ideally, the support provides mechanical support to permit the imposition of a pressure difference across the membrane without damaging it but does not pose significant resistance to permeation. Membrane permeation rates are controlled by the material comprising the dense layer, and the value of *d* in Equation (1) is equal to the effective thickness of this layer. Direct measurement of the layer thickness is often difficult. However, the value can be estimated from permeation rates measured for both the supported membrane and thicker samples of a dense unsupported membrane.

The ability of a membrane to separate two gases is given by its selectivity, which is defined for a pair of gases, *i* and *j*, as the ratio of gas permeabilities or permeances:(2)$$\alpha_{i,j}=\frac{Q_{i}}{Q_{j}}=\frac{q_{i}}{q_{j}}$$
where component *i* is the gas with the higher permeability, resulting in a selectivity greater than one. Selectivity and permeability determine the economics of a gas separation process [5]. Selectivity controls the energy (operating) cost, while permeability controls the capital (membrane area) cost. Increasing selectivity reduces the amount of gas that must permeate from the high feed pressure to the low permeate pressure to achieve product purity targets; this reduces the compression energy lost due to permeation. Increasing permeability or permeance, the gas permeation flux per unit driving force, reduces the membrane area and associated capital cost required to achieve a target feed or product flow rate.

A trade-off exists between permeability and selectivity for a given gas pair, as illustrated in Figure 2. The permeability–selectivity combination of polymeric membrane materials reported in the literature lies below a line commonly referred to as the Robeson upper bound [11]. This observation suggests that a limit exists on the ability to increase selectivity and permeability simultaneously through changes in polymer architecture. Recent developments in membrane materials such as facilitated transport, mixed matrix, and molecular sieve membranes indicate that the upper bound can be surpassed through the addition of non-polymeric components and introduction of different transport mechanisms [3,7].

## 3. Membrane Module Designs

Large membrane areas are required for industrial gas separations to process desired gas flow rates. Membranes are packaged into modular units to form compact, high-surface-area-per-unit-volume contactors. These modules are typically combined in parallel or series inside a pressure vessel or case for use in a separation process. Three primary module designs have been developed: plate and frame, spiral wound, and shell and tube. Flat-sheet membranes are used in the plate-and-frame and spiral-wound designs, while hollow cylindrical membrane fibers are used in the shell-and-tube design [5].

### 3.1. Plate-and-Frame Module

Plate-and-frame modules consist of stacks of membrane sheets. Flow channels are created between adjacent membranes with a spacer. The spacers also maintain the flow channel when a pressure difference is imposed across the membrane to drive gas permeation and can mix the fluid in the flow channel to reduce concentration polarization. The spacers for the feed and permeate channels commonly are different since a finer mesh is needed to support the membrane and prevent rupture in the lower pressure gas permeate channel because the membrane is pressed into the spacer by the applied pressure difference.

Membranes are glued together along the edges such that, when the module is placed in a case, gas streams can be introduced into and removed from alternating channels through external connections, as illustrated in Figure 3. This leads to crossflow contacting where the permeate flows in the normal direction to the feed. Restricting where the gas stream enters along the edges can introduce partial countercurrent contacting (not illustrated). Such modules are one of the oldest membrane systems and are often used in dead-end filtration. Capital costs are usually higher for plate-and-frame modules, but operational costs are typically lower due to lower pressure drops [5,12].

### 3.2. Spiral-Wound Module

Flat-sheet membranes also can be used to fabricate spiral-wound modules. Like plate-and-frame modules, membrane sheets are separated by feed and permeate spacers, and adjacent membranes are glued to allow gas introduction and removal from separate flow channels. In contrast to plate-and-frame modules, long membrane sheets are used. One long membrane is glued along the long edges of the sheet, and one short edge with a permeate spacer and central permeate collection tube on top to create a “leaf”, as illustrated in Figure 4. The permeate collection tube is rolled to wrap the leaf around the tube and form the module. Multiple leafs can be placed in a single module to increase the total membrane area, while reducing leaf length to minimize the permeate pressure drop inside each leaf. The module is placed into a case with external connections that allow for the introduction and removal of the feed and reject along the leaf exterior and separate permeate collection from the leaf interior from a central tube.

The leaf glue lines and connection to the permeate collection tube create crossflow contacting in the module. The feed flows parallel to the permeate collection tube from one face of the module to the other, while the permeate flows perpendicular to the feed in a spiral fashion through the leaf to the permeate collection tube. Compared to plate-and-frame modules, spiral-wound modules can offer greater area per unit volume and more facile manufacture and module handling [5,12].

### 3.3. Shell-and-Tube Module

Membranes in the form of hollow fibers or tubes are commonly formed into modules like the one shown in Figure 5. The ends of the bundle are enclosed in a tube-sheet material that seals the fiber together. The tube sheet is machined such that the fiber lumens (interior) are open, and the tube sheet can be sealed to the interior of an external cylindrical case. External ports on the end of the case allow fluid to be introduced to and removed from the fiber lumens, while ports on the circumference allow for separate fluid introduction to and removal from the shell (the space outside the fibers). Such a configuration is the mass transfer equivalent of a shell-and-tube heat exchanger and provides nominal countercurrent contacting with the potential to use a sweep stream that can dilute the permeate and thereby enhance permeation rates [5,12]. Larger hollow-fiber membranes, fibers with an outer diameter greater than ~0.5 cm, often are referred to as tubular or capillary membranes, but the module design is the same as for finer hollow fibers.

## 4. Module Flow Patterns

Figure 6 illustrates the three primary contacting configurations considered in module design and performance calculations [5,12]:Co-current: feed and permeate flow parallel to each other in the same direction.Countercurrent: feed and permeate flow parallel to each other in opposite directions.Crossflow: feed and permeate flow perpendicular to each other.

**Figure 6 membranes-13-00639-f006:**
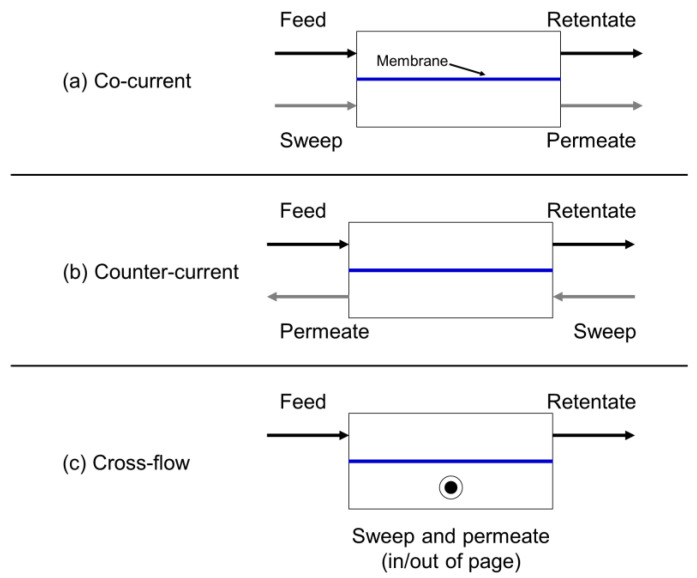
Membrane module flow configurations (with sweep): (**a**) co-current, (**b**) countercurrent, and (**c**) crossflow.

All three of these flow configurations can be achieved nominally in either a flat-sheet or hollow-fiber module. Additionally, all three can use a sweep stream on the permeate side of the membrane to increase the partial pressure driving force for permeation.

For ideal contacting conditions (i.e., in the absence of significant pressure drop and other module inefficiencies arising from non-uniform membrane properties and fluid distribution, as discussed later), countercurrent contacting maximizes partial pressure differences within the module and yields the best performance, as measured by module productivity (product flow rate per unit membrane area) and gas recovery (fraction of feed recovered as the desired product). However, the differences between the configurations may be small depending on the gas separation and the module stage cut (i.e., the fraction of the feed that permeates). As the stage cut decreases, concentration changes along the feed and permeate channels decrease, and performance differences between the configurations become smaller.

Membrane and module selection is a critical task when developing a new membrane gas separation process. The choice of flat-sheet or hollow-fiber membrane occurs first and dictates what module options exist. If either flat-sheet or hollow-fiber membrane modules offered inherently superior economic performance, manufacturers would opt to produce that type of module. However, since all three contacting configurations are nominally possible with both flat-sheet and hollow-fiber membranes, the choice is based primarily on membrane- and module-manufacturing expertise. The production of high-performance gas separation membranes requires creating an effective dense separation layer less than ~0.1 micron thick on a porous support that provides the requisite strength to withstand the pressure difference across the membrane but does not pose a resistance to gas permeation. The art and science of membrane formation are highly specific to the membrane type and closely guarded by manufacturers, so the development of new membranes typically relies on adapting an existing process.

While similar contacting configurations are available in flat-sheet and hollow-fiber modules, differences between the module types exist and are summarized in Table 1 [5,12]. Cost estimates of module manufacture vary widely and depend strongly on the degree of automation. The patent literature describes automated processes for the formation of large fiber bundles [14], but the automated manufacture of large flat-sheet modules has lagged.

The ease of countercurrent contacting is greater in hollow-fiber modules than flat-sheet modules, due to differences in module geometry, while concentration polarization is more difficult to mitigate in hollow-fiber modules, especially in the fiber lumens which lack spacers to promote fluid mixing. For many current commercial gas separations, concentration polarization is not significant, and this would favor the shell-and-tube design. The pressure drop in both module types can be reduced by increasing the flow channel dimensions. Increasing the fiber size or spacer thickness will lead to lower pressure drops but will also reduce the membrane area per unit module volume. With shell-and-tube designs, pressure drops in the shell are lower than in the lumen which may dictate where the feed is introduced. Additionally, shell feed is preferred for a higher-pressure operation since hollow fibers are stronger when externally pressurized.

## 5. Membrane Modeling Review

A simulation of membrane-based gas separation systems requires the development of stable and robust models that can efficiently predict the separation performance for different flow configurations and module geometries. Mathematical modeling of gas membrane modules was first introduced by Weller and Steiner [15] in 1950, and their work has served as the basis for future model development. Shindo et al. [16] developed a calculation methodology for all five flow configurations for multicomponent gas separations. Pan [17,18] proposed solving module performance models with numerical integration of the governing differential equation mass balances, using split boundary conditions for the retentate and permeate side. However, this approach can be highly sensitive to the initial guess and computationally expensive when refining the solution to reduce numerical approximation errors. Khalipour et al. [19] extended Pan’s model into a system of backward finite differential equations that are solved using a Gauss–Seidel algorithm for an isothermal co-current and countercurrent module. Chowdhury et al. [20] reformulated Pan’s model as an initial value problem in which either an Adams–Moulton’s or Gear’s backward differentiation method is used for solving the nonlinear differential equations. Kovvali et al. [21] simplified Pan’s model as a set of nonlinear equations by assuming a linear relationship between the permeate and inlet stream compositions; this approach can reduce the computational effort while maintaining an acceptable accuracy. Sengupta and Sirkar [22] provide an excellent review of this early work and the numerical methods used to obtain numerical approximations to the governing equations.

Coker et al. [23,24] developed a stage-based approach to convert the differential mass balance equations into a set of nonlinear algebraic equations by assuming that the module consists of a series of well-mixed stages. The resulting equations are solved iteratively using a direct substitution algorithm that requires solution of a set of linear equations each iteration. The coefficients of the linear equations form a block tridiagonal matrix that allows for an efficient simultaneous solution using the Thomas Algorithm. The principal drawback of this approach is the sensitivity of the convergence process on the initial solution guess; the use of the solution for an equivalent crossflow module is recommended. This methodology is readily implemented in different computational environments and allows for a facile addition of non-idealities within the membrane module, such as pressure drop or temperature variation effects that arise due to the expansion-driven process.

The incorporation of the membrane numerical solving algorithm into a commercial process simulator such as Aspen Plus allows the user to understand how the model will behave under a range of operating conditions when connected to other process units. Commercial process simulators often do not have a membrane separation unit as part of their unit operations library; therefore, the implementation of a membrane module in a simulator such as Aspen Plus can help industry and academia further the research and development of membrane gas separation processes.

The inclusion of robust mathematical membrane models into commercial process simulators is well documented in the literature. However, some of these models are found to be simplistic due to the assumptions that are made with respect to the physical, geometrical and transport properties of the process. The baseline approach found in academic literature for membrane modeling consists of assuming the following conditions [15,16]:Steady state;Ideal gas behavior;Isothermal conditions;Constant permeance;Constant pressure in feed/retentate and permeate channels;No axial mixing;Uniform flow channel size;Negligible concentration polarization effects;Laminar flow.

Under these assumptions, the model for an “ideal” module consists of a set of differential mass balances that describe the concentration profiles of each component along the length of the module. Following Coker et al. [23], these equations can be solved by dividing the module length into a series of *N* well-mixed, as illustrated in Figure 7.

The overall mass balances for an individual stage, as shown in Figure 8, are given by Equations (3a)–(3c) for the countercurrent, crossflow, and co-current flow, respectively:(3a)$${R_{f}}_{\left(j-1\right)}+{P_{f}}_{\left(j+1\right)}={P_{f}}_{\left(j\right)}+{R_{f}}_{\left(j\right)}$$
(3b)$${R_{f}}_{\left(j-1\right)}={P_{f}}_{\left(j\right)}+{R_{f}}_{\left(j\right)}$$
(3c)$${R_{f}}_{\left(j-1\right)}+{P_{f}}_{\left(j-1\right)}={P_{f}}_{\left(j\right)}+{R_{f}}_{\left(j\right)}$$
where *R_f_* and *P_f_* are the retentate and permeate flows, respectively; and the subscript indicates the stage the flow comes from. The component flow rates can be calculated similarly by using Equations (4a)–(4c) for the countercurrent, crossflow, and co-current flow, respectively:(4a)$${R_{f}}_{\left(j-1\right)}{x_{r}}_{\left(j-1,i\right)}+{P_{f}}_{\left(j+1\right)}{y_{p}}_{\left(j+1,i\right)}={\;P_{f}}_{\left(j\right)}{y_{p}}_{\left(j,i\right)}+{R_{f}}_{\left(j\right)}{x_{r}}_{\left(j,i\right)}$$
(4b)$${R_{f}}_{\left(j-1\right)}{x_{r}}_{\left(j-1,i\right)}={\;P_{f}}_{\left(j\right)}{y_{p}}_{\left(j,i\right)}+{R_{f}}_{\left(j\right)}{x_{r}}_{\left(j,i\right)}$$
(4c)$${R_{f}}_{\left(j-1\right)}{x_{r}}_{\left(j-1,i\right)}+{P_{f}}_{\left(j-1\right)}{y_{p}}_{\left(j-1,i\right)}={\;P_{f}}_{\left(j\right)}{y_{p}}_{\left(j,i\right)}+{R_{f}}_{\left(j\right)}{x_{r}}_{\left(j,i\right)}$$
where *x_r_* and *y_p_* are the mole fractions of component i in the retentate and permeate flows, respectively; and the subscripts *j* − 1, *j*, and *j* + 1 indicate the stage that the flow comes from.

Permeation across the membrane is calculated from Equation (1), using Equations (5a)–(5c) for countercurrent, crossflow, and co-current flow, respectively:(5a)$${\;P_{f}}_{\left(j\right)}{y_{p}}_{\left(j,i\right)}-{P_{f}}_{\left(j+1\right)}{y_{p}}_{\left(j+1,i\right)}=q_{\left(j,i\right)}A(P_{r_{\left(j\right)}}{x_{r}}_{\left(j,i\right)}-P_{p_{\left(j\right)}}{y_{p}}_{\left(j,i\right)})$$
(5b)$${\;P_{f}}_{\left(j\right)}{y_{p}}_{\left(j,i\right)}=q_{\left(j,i\right)}A(P_{r_{\left(j\right)}}{x_{r}}_{\left(j,i\right)}-P_{p_{\left(j\right)}}{y_{p}}_{\left(j,i\right)})$$
(5c)$${\;P_{f}}_{\left(j\right)}{y_{p}}_{\left(j,i\right)}-{P_{f}}_{\left(j-1\right)}{y_{p}}_{\left(j-1,i\right)}=q_{\left(j,i\right)}A(P_{r_{\left(j\right)}}{x_{r}}_{\left(j,i\right)}-P_{p_{\left(j\right)}}{y_{p}}_{\left(j,i\right)})$$
where $P_{p_{\left(j\right)}}$ and $P_{r_{\left(j\right)}}$ are the permeate and retentate pressures, respectively, in stage *j*. The retentate and permeate mole fractions must sum to unity:(6)$$\sum_{i=1}^{nc}{x_{r}}_{\left(j,i\right)}=1$$
(7)$$\sum_{i=1}^{nc}{y_{p}}_{\left(j,i\right)}=1$$
where nc represents the number of components in the gas permeation process.

In the case of different module geometries, the active membrane area (*A*) for which permeation occurs is given by Equation (8) for hollow-fiber membranes based on the outer fiber diameter, and by Equation (9) for flat-sheet and spiral-wound membranes:(8)$$A=\frac{\pi ODLN_{f}}{N}$$
(9)$$A=\frac{W\times L\times N_{sheets}}{N}$$
where L is the active permeating length of the membrane (excluding regions included in either tubesheets or glue lines), OD is the fiber outside diameter, $N_{f}$ is the number of fibers within the module, W is the width, $N_{sheets}$ is the number of membrane sheets, and N is the number of discretization stages. *N* must be increased until the performance predictions become independent of *N*.

## 6. Modeling Non-Idealities in Membrane Gas Separation Modules

Various non-ideal effects are observed during the operation of gas separation membrane modules. These non-ideal effects can have adverse effects on gas separation performance, and, thus, these effects should be included in models to improve accuracy if possible. Table 2 summarizes models reported in the literature that have accounted for the following non-ideal effects:Real gas behavior;Pressure drop in both permeate and retentate channels;Non-isothermal behavior;Concentration polarization;Variable permeance;Non-uniform membrane properties.

**Table 2 membranes-13-00639-t002:** Summary of available models in the literature that model the following non-ideal behaviors: (1) real gas behavior, (2) channel pressure drop, (3) non-isothermal behavior, (4) concentration polarization, (5) variable permeance, and (6) non-uniform membrane properties. Commercial software packages are also listed for models that have been integrated into process simulation software.

References	Non-Ideal Conditions Modeled						Commercial Software Integration
	1	2	3	4	5	6	
Pan [18]	-	Lumen	-	-	-	-	-
Khalipour et al. [19]	-	Lumen	-	-	-	-	-
Chowdhury et al. [20]	-	Lumen and shell	-	-	-	-	Aspen Plus
Kovvali et al. 1994 [21]	-	Lumen and shell	-	-	-	-	-
Coker et al. [23]	-	Lumen and shell	+	-	+	-	-
Feng et al. [25]	-	Lumen	-	+	-	-	-
Rautenbach et al. [26]	-	Lumen	-	-	-	+	-
Scholz et al. [27]	+	Lumen and shell	+	+	+++	-	ACM
Miandoab et al. [28,29]	+	Lumen and shell	+	+	+++	-	ACM
Che Mat et al. [30]	+	Lumen and shell	+	-	+	-	Aspen Plus
Bounaceour et al. [31]	-	-	-	-	+++	-	Aspen Plus
Sonalkar et al. [32]	-	Lumen	-	-	-	+	-
Ahmad et al. [33]	-	Lumen	+	-	++	-	Aspen HYSIS
Chu et al. [34]	-	Lumen and shell	-	-	-	-	-
Brinkmann et al. [35]	+	Both channels	+	+	+++	-	ACM
Mourgues et al. [36]	-	-	-	+	-	-	-
Hensen [37]	-	Lumen and shell	-	-	-	-	ACM and gPROMS
Marriott et al. [38]	-	Lumen and shell	+	-	-	-	gPROMS
Tessendorf et al. [39]	+	Lumen and shell	-	-	-	-	OPTISIM
DeJaco et al. [40]	+	Both channels	-	-	-	-	-
Qi et al. [41]	-	Permeate channel	-	-	-	-	-
Aiman et al. [42]	-	Permeate channel	-	-	-	-	-
Rivero et al. [43]	-	Both channels	-	-	-	-	-

‘-’ indicates that the model did not account for that non-ideal effect, ‘+’ indicates that the model did account for that non-ideal effect. For column ‘5′, ‘+’ indicates that permeance was modeled as a function of temperature only; ‘++’ indicates that permeance was modeled as a function of temperature and pressure; and ‘+++’ indicates that permeance was modeled as a function of temperature, pressure, and composition.

The impacts of each non-ideality on improving performance predictions and module design are discussed. In all cases, including the non-ideality either led to an improvement in predictions or highlighted a key variable for consideration in future module designs. While most studies have been limited to countercurrent modules, the effect of each non-ideality would be included for crossflow and co-current flow following the approach for countercurrent flows.

### 6.1. Real Gas Behavior

Membrane gas separations can be carried out at high feed pressures (>10 bar), and therefore ideal gas behavior may not be accurate. This can affect the driving force for gas permeation and requires the researcher to replace the partial pressure driving force for permeation with a fugacity driving force. The Soave–Redlich–Kwong equation of state (EOS) has proven to provide a good description of the pressure–volume–temperature (PVT) properties for many industrially relevant gas mixtures [28,30]. A study conducted by Scholz et al. [27] for biogas separation concluded that real gas effects should be accounted for when operating at pressures exceeding 10 bar. The fugacities were calculated from the EOS and used to calculate the gas permeation rates. The availability of this equation of state is common in well-known process simulators such as Aspen Plus [37].

### 6.2. Friction Losses

Pressure drops are expected to occur in all module geometries in both the retentate and permeate channels, reducing the driving force for separation. A pressure drop also increases the energy requirement for the separation process. This is especially important in CO_2_-capture processes where large volumes of gas must be pushed through membranes, leading to large operating costs if pressure drops are high [44]. Pressure drops through each module will be different due to their inherent geometries. Therefore, module selection is an important aspect during process design.

Pressure drop in hollow-fiber modules is described by the Hagen–Poiseuille equation for laminar flow in both the lumen and shell side (assuming an equivalent hydraulic diameter for the shell domain). The Hagen–Poiseuille equation has been studied extensively in hollow-fiber membrane modules for different gas separations and provides a good description of experimental data [45]. For a hollow-fiber membrane module, pressure drops per stage are given by the following calculations [24,28]:(10)$$∆P_{l}=\frac{128\mu_{r}L_{f}}{\pi N_{f}{ID}^{4}\rho_{r}}\times \frac{L}{N}$$
(11)$$∆P_{s}=\frac{128\mu_{p}S_{f}}{\pi \rho_{p}{d_{hyd}}^{2}({d_{m}}^{2}-Nf{OD}^{2})}\times \frac{L}{N}$$
where μ is the retentate (*r*) or permeate (*p*) viscosity; ρ is the retentate or permeate density; ∆P is the pressure drop of the lumen (*l*) or shell (*s*) side; ID is the internal diameter of the fiber; $N_{f}$ is the number of fibers; L is the fiber length; $d_{m}$ is the module diameter; OD is the fiber outside diameter; N is the number of stages; and $L_{f}$ and $S_{f}$ are the lumen and shell flow rate, respectively.

Lumen-side pressure drops can be detrimental to performance when designing a module with small fiber diameters, whereas shell-side pressure drops can be significant at high packing densities. Normally, pressure drops through the shell side are much smaller than those on the lumen side due to limitations on the module packing density required to introduce and remove gas through external ports on the case from the shell. Chu et al. [34] demonstrated that, for natural gas separation, pressure drops on the lumen side can lead to significant methane-retentate recovery loss, thereby increasing the pressure requirement needed for separation.

Pressure drop in flat-sheet and spiral-wound membrane modules can be approximated by the Darcy–Weisbach law expression [35,46]:(12)$$∆P=\frac{\lambda \rho v^{2}}{2d_{h}}\times \frac{L_{c}}{N}$$
where λ is the friction factor, ρ is gas density, $d_{h}$ is the hydraulic diameter, $L_{c}$ is the channel length, and v is the bulk velocity.

When contrasting spiral-wound and flat-sheet sweep pressure drops for a 20 TPD CO_2_ capture multistage membrane system, the flat-sheet membrane displayed a four-times-lower pressure drop (<1 psia) than the spiral-wound modules [47]. However, this difference comes at the expense of a lower packing density and, hence, larger module footprint.

### 6.3. Non-Isothermal Behavior

Gas permeation through a membrane is analogous to the isenthalpic expansion of a gas through a throttling valve. Therefore, as the gas permeates, changes in temperature along the length of the membrane can be expected according to the Joule–Thomson coefficients for each component. Temperature changes due to the Joule–Thomson effect impact the permeance and selectivity of the membrane due to the following Arrhenius equation for the temperature dependence of gas permeability:(13)$${Q(T)}_{\left(j,i\right)}=Q_{i}exp\left(\frac{-E_{a,i}}{R}\left(\frac{1}{T_{r,j}}-\frac{1}{T_{0}}\right)\right)$$
where $E_{a,i}$ is the activation energy for component *i*, R is the gas constant (8.314 J/mol/K), $T_{r}$ is the retentate temperature at each stage j, $T_{0}$ is the reference temperature, $Q_{i}$ is the temperature independent pre-exponential factor, and ${Q(T)}_{\left(j,i\right)}$ is the temperature dependent permeability of component i at stage j. Note that the resistance to heat transfer across the membrane is small, so the temperature differences between permeate and retentate are often small.

Coker et al. [23] studied non-isothermal behavior for binary and multicomponent gas mixtures in a hollow-fiber membrane module. The proposed model was validated with experimental data and demonstrated that, for multicomponent natural gas separation, the temperature can decrease up to 40 °C at the 50% stage cut, and increasing the CO_2_ feed stream composition led to greater temperature changes due to the high Joule–Thomson coefficient for CO_2_. Ahmad et al. [33] also demonstrated that the permeate gas temperature decreases as the CO_2_ feed content increases for higher stage cuts in a non-isothermal natural gas separation experiment. However, heating of the retentate stream can occur when separating a CO_2_/H_2_ stream from pre-combustion power plants, as shown by Miandoab et al. [28], due to the negative Joule–Thomson coefficient for hydrogen. Temperature-change effects are most critical to model for gas separations with gases that possess large Joule–Thomson coefficients, especially at high stage cuts and high pressure ratios [28,31,33,35].

### 6.4. Concentration Polarization

Concentration polarization influences the mass transfer process by slowing the flow of the most permeable component, while increasing the flow of the least permeable gas. Therefore, the concentration of the components changes from the boundary layer surface to inside the porous support and membrane surface. Polarization effects have been observed only for high-flux membranes and negatively affect module performance by decreasing product purity most prominently at high stage cuts [25].

Concentration polarization can be modeled by introducing a mass transfer coefficient to describe the boundary layers on either side of the membrane and the resistance of the support [28,29]. The dependence of the mass transfer coefficient on the module geometry and operating conditions is complex and not well understood.

Pan [17] noted the potential impact of concentration polarization in the support of a high-flux membrane and demonstrated that it can lead to effective crossflow contacting even if the module is operated nominally in countercurrent mode. Sidhoum et al. [48] showed that if the support resistance and contacting-gas-phase resistances are sufficiently low, the performance of asymmetric hollow-fiber membrane modules, with the membrane discriminating layer on the fiber outer surface, does not depend on whether the module is lumen or shell fed. Thus, proper design of the support is critical to minimize internal concentration polarization and maximize module performance.

Alpers et al. [49] documented the importance of external gas-phase concentration polarization for organic vapor separations with high-flux membranes. The apparent membrane selectivity increased dramatically with the flow rate, and the changes could be described quantitatively with an appropriate mass transfer coefficient correlation for the contacting gas phases.

Miandoab et al. [28] studied the impact of concentration polarization in a hollow-fiber membrane for pre-combustion CO_2_ capture. Concentration polarization had less of an impact on CO_2_ and H_2_ selective membranes than other non-ideal effects. Such a conclusion concurs with the work of Mourgues et al. [36] in which concentration polarization was prominent only when component permeance exceeded 1000 GPU and selectivity was greater than 100. Scholz et al. [27] performed a similar study for assessing the impact of concentration polarization for biogas upgrading and concluded that concentration polarization can be neglected for low-flux membranes. Furthermore, Feng et al. [25] included the influence of concentration polarization for air separation in lumen-fed hollow-fiber modules and found that concentration polarization may become significant at high stage cuts.

### 6.5. Non-Uniform Membrane and Module Properties

Membrane module defects that arise from manufacturing practices can have detrimental effects in terms of the separation performance. In hollow-fiber membrane modules, inherent variations in fiber properties, such as permeance and internal diameter, can induce flow maldistribution effects that can dramatically affect module performance and even limit the product purity that can be achieved [26]. Similar effects can occur in flat-sheet and spiral-wound modules, where variations in permeate channel height and membrane permeance can diminish the recovery for a given retentate product and thereby increase the membrane area required for separation [26,32].

Sonalkar et al. [32] provided a detailed framework for studying the impact of fiber variability on module performance. The model assumes a Gaussian distribution for fiber internal diameter, selectivity, and gas permeance. A lumen-fed hollow-fiber membrane module was considered with perfect permeate mixing and no permeate mixing, and in the former, the permeate from all fibers was well mixed along the length of the module, whereas in the latter, no mixing between the fiber permeate took place. Simulation results, which were validated against experimental data from an air separation module, show that variation in inner diameter (ID) has the greatest effect on module performance as compared to varying the permeance or selectivity of the fibers.

As ID variation increases, recovery decreases. This inverse relationship is due to the dependence of flowrate on the fourth power of the inner diameter, which is given by the Hagen–Poiseuille equation. For instance, a 20% inner diameter variation can decrease the retentate recovery by as much as 30% of the original value in the case of binary air separation. Variations in fiber permeance have a moderate impact on performance compared to ID variation; a permeance variation of 30% is analogous to a 10% ID variation.

An additional study, following similar methodology, investigated the impact of channel height variation on flat-sheet membrane and plate-and-frame module performance [43]. The model shows that, as channel variability increases, recovery decreases, as the flowrate is dependent on the channel height, but to the third power. Since the dependency is smaller than a hollow-fiber membrane, the effect of flow channel variability is slightly less significant for plate-and-frame modules compared to hollow-fiber modules. However, the study demonstrates that both membrane configurations see performance decline with increasing variability, allowing for further investigations into other non-uniform conditions.

### 6.6. Competitive Sorption, Penetrant Blocking, and Plasticization Effects

The presence of highly condensable gases in glassy polymeric membranes can induce plasticization effects that can negatively impact the physical properties of the membrane, leading to a reduction in the gas permeance and selectivity [50]. Plasticization effects lead to enhanced diffusion of the components due to the increased segmental motion stemming from the presence of highly sorbent condensable gases such as CO_2_ and CH_4_. Furthermore, competitive sorption effects and penetrant blocking can occur due to the excess free volume present in glassy polymers; these three phenomena can be mathematically described by the dual sorption/partial immobilization model [51,52]. Visser et al. [50] conducted mixed gas permeation experiments for a CO_2_/N_2_ mixture and showed that plasticization effects dominate at high pressures and low concentrations of the inert gas, whereas competitive sorption effects become stronger at higher inert gas concentrations. A subtle balance exists between plasticization and competitive sorption, as the latter can counterbalance the effects induced by plasticization at a high concentration of the inert gas.

Commonly, permeance is assumed to be constant or temperature independent in most models; therefore, the inclusion of a pressure-, temperature-, and composition-dependent permeance is necessary to fully characterize membrane permeation. Miandoab et al. [29] presented a comprehensive permeance model for investigating these permeance dependencies based on a fugacity-dependent permeance for glassy polymers that includes membrane plasticization, penetrant blocking, and competitive sorption effects [51,53,54]. The proposed model was validated with experimental permeance measurements and used for a sensitivity analysis in biogas upgrading. The results showed that competitive sorption and plasticization by CO_2_- and H_2_O-induced free volume blocking can have a significant effect. Additionally, a humidified gas feed showed a larger decrease in performance as compared to a dry gas stream due to the induced free volume blocking by water vapor and competitive sorption effects. Scholz et al. [27] performed a similar study of biogas upgrading by using a second-order polynomial to express the pressure- and composition-dependent permeance. Although the model predicted more pronounced effects when considering competitive sorption and plasticization effects, the method employed was not an adequate description of permeation physics. Bounaceur and Ahmad et al. [31,33] performed similar studies; however, the former did not consider the effect of composition, whereas the latter assumed constant gas diffusivity. The influence of non-ideal effects due to membrane permeance combined with those of module operation and module fabrication defects should be taken into consideration to accurately quantify the overall module performance.

## 7. Modeling Applications and Challenges

### 7.1. Dimensional Modeling

One-dimensional membrane module modeling based on an ideal contacting configuration is common in the literature. Axial variations in composition, pressure, and temperature are predicted by assuming plug flow behavior. In the case of hollow-fiber membrane modules with counter and co-current flow configurations, the one-dimensional assumption often allows researchers to make accurate predictions of experimental data since the flow is parallel to the membrane [17,24,25,39]. However, in the case of crossflow hollow-fiber membrane modules, variations in both the axial and radial direction can be expected due to concentration changes and pressure buildup inside the module; this behavior is analogous to that of spiral-wound and flat-sheet membrane modules in which the permeate flows perpendicular to the feed stream. DeJaco et al. [40] proposed a 1D and 2D model for simulating air separation from spiral-wound membrane modules. His results showed that the 1D model provided a good approximation to the results of the more detailed 2D model over a range of stage cuts and oxygen permeate concentrations that were fit with experimental data. Additionally, the 1D spatial distribution of momentum variables (pressure drop and velocity) was in good agreement with the one-dimensional averaged variables from the 2D model.

Brinkmann et al. [35] developed a 1D model for envelope-type and flat-sheet modules and a 2D model for spiral-wound membrane modules in which non-ideal effects were considered for all modules studied. The simulation results from the modules investigated were in good agreement with the pilot plant experiments conducted by the author. Similarly, Dias et al. [55] presented a 1D and 2D model for simulating the performance of a spiral-wound membrane module under crossflow operation. The 2D profiles obtained were in better agreement with the literature data as compared to the 1D model.

A 3D computational fluid dynamics model is expected to provide enhanced understanding of the flow and mass transfer behavior inside membrane modules, but implementing this level of modeling into a commercial process simulator becomes unpractical due to higher computational times. Moreover, Haddadi et al. [56] performed a study comparing a 1D model versus a 3D model for a hollow-fiber membrane module and concluded that both models showed good agreement with the experimental data, with less than 2% error. These results suggest that 1D models may provide good performance predictions for initial process design, but higher dimensional modules will be required to improve predictions and agreement with experimental data.

### 7.2. Software Implementation

Software selection for membrane module modeling is a key step for model development. The decision to model a membrane module depends on the spatial dimension used and the level of complexity of the model equations. Several works in the literature [27,28,29,37] have used Aspen Custom Modeler (ACM), which is an equation-oriented modeling software that has built-in numerical solvers that are capable of solving linear, nonlinear, ordinary, and partial differential equations. The customized unit operation model can be interfaced with the Aspen properties package and easily exported to Aspen Plus and HYSYS for performing flowsheet simulations. ACM offers a unique advantage over other software tools, such as MATLAB, Excel VBA, and others, because there is no need for coding the numerical method or algorithm for solving these equations, and the interfacing process with a commercial process simulator is simpler.

Additional software tools such as gPROMS have been used by Hensen [37] and Marriott et al. [38] to model hollow-fiber membrane modules under different operating conditions; the model can also be exported to Aspen Plus through CAPE-OPEN (Computer-Aided Process Engineering), which allows for the interoperability between a process modeling environment and a custom process modeling component. The incorporation of customized models in process simulators also can be performed through user-defined subroutines that are available in the program (Aspen Plus, HYSYS). However, this requires the user to code the membrane in FORTRAN, Visual Basic, C, or C++ programming languages and interface the routines with the process simulator.

If a more sophisticated modeling approach is desired for analyzing in detail the mass, momentum, and heat-transfer effects inside the membrane module, 3D software tools such as COMSOL Multiphysics, ANSYS, and OpenFoam [56,57,58] can be used to account for actual module geometry and associated fluid contacting. However, 3D models can be computationally expensive, and therefore simulation times must be balanced with the level of accuracy wanted as the meshing requirements required for the solution to be mesh independent will lead to longer run times.

### 7.3. Comparisons with Pilot-Scale Experiments

A great majority of the developed models for membrane gas separations are validated against experimental data from small-scale applications. There is a limited amount of work performed with model validation against pilot-plant-scale experiments. Brinkman et al. [35,59] developed carbon capture membrane models using the Aspen Custom Modeler (ACM) and compared them against experimental data from several field trials completed at three different locations in Germany. The operating conditions and design variables differ from each test conducted, with the purpose of evaluating the performance of the membrane module under a wide range of flows, feed compositions, and other parameters. The pilot-plant experiments were completed in a steady-state fashion over a period of two months for the first two trials and 400 h for the third one. The results from the simulation models were in good agreement with the pilot-plant data; however, the prediction was slightly poorer when concentration polarization effects became prominent at low feed-flow rates. Furthermore, high feed pressures increase the model discrepancies due to the complex behavior of the membrane permeance that was not captured by the model. A decrease in performance was not observed for the period of operation, and thus, the experiments conducted could be predicted without accounting for membrane aging.

Choi et al. [60] completed a pilot-scale membrane plant study for the separation of CO_2_ from liquefied natural fired flue gas. A hollow-fiber simulation was employed to establish comparisons against the experimental data. The pilot plant in this study consists of a multistage membrane separation process with feed gas and vacuum compression. The process was operated under different operating scenarios, and a good agreement between the numerical simulation and the field data was found. However, when atmospheric pressure was applied on the permeate side, the simulations provided poorer predictions of the required membrane area.

Fixed-site carrier membranes developed by Norwegian University of Science and Technology (NTNU) in Norway were pilot tested under different environments and module configurations for CO_2_ capture. Sandru et al. [61] developed plate-and-frame modules (0.25 m^2^–1.5 m^2^) that were tested for more than six months in a coal-fired power plant in Portugal. The separation performance was consistent over the testing period and showed a good dynamic response to process upsets. However, the flat-sheet membranes were not efficient and were difficult to scale up. Therefore, Hagg et al. [62] designed a hollow-fiber membrane module (4 m^2^) for testing in a cement plant. The module performed well and consistently under large testing periods and harsh conditions. He et al. [63] scaled up the hollow-fiber membranes (4.0 m^2^–10 m^2^) in order to find the best operating conditions and improve the module’s performance. The membranes were subjected to several experimental runs in which multiple process parameters were varied to assess the membrane’s performance and behavior. The results from this testing campaign indicate that a bore-side-fed module provided better gas distribution within the module and, thus, improved the module’s efficiency compared to a shell-fed module. NTNU demonstrated the compactness and robustness of their membrane modules at a pilot-plant level, but they did not use uncertainty quantification within a process model to account for measurement error and bias while conducting model validation.

Facilitated transport membranes comprising polyvinylamine as a fixed carrier and an amino acid salt as a mobile carrier developed by The Ohio State University [64,65,66] also offer attractive performance for CO_2_ capture. Transport models were developed and validated for the membrane with data from test coupons, and the model was used to perform technoeconomic analyses. Notably, spiral-wound modules were tested at the National Carbon Capture Center, but detailed comparisons to module simulations were not provided.

An uncertainty quantification approach was taken by DeJaco et al. [40] to help validate experimental data from an air-separation spiral-wound membrane module with simulation results. Uncertainties in seven input model parameters were evaluated in the simulation and propagated to the output variables: stage cut and permeate mole fraction. The results indicate that the uncertainty for both calculated and experimental stage cuts is predominant at low feed flows due to measurement limitations. Uncertainty in the permeate purity was found to increase when operating the module at high feed pressures; therefore, more precise measurements of component permeances are needed to accurately determine the change in permeate purity.

There has been a large quantity of bench-scale/pilot-plan-scale tests reported in the literature for different gas separations with several module types, as shown in Table 3. However, only a few of these experimental tests were validated with membrane models, and only DeJaco et al. [40] accounted for rigorous parametric uncertainty of the process model.

The process models for membrane-based separation processes are predominantly deterministic in nature—rigorous parametric uncertainty quantification has not been accounted for. Sensitivity studies must be performed to assess the impact of errors in process variable measurements and help identify input parameters that have the largest contribution to the output variable of interest.

Additionally, optimization studies have been completed to identify regions where the membrane process can meet desired product purity and recovery targets. Propagating the uncertainty through the process model can allow for the estimation of uncertainty in meeting these targets. It has been proven that when taking parameter uncertainty into consideration, the model is able to accurately predict large-scale pilot-plant data for CO_2_ capture in solvent systems [74]. Membrane-based CO_2_-capture technologies can clearly benefit from the implementation of uncertainty quantification into process models that can help replicate the pilot-plant data while maximizing the knowledge in the stochastic modeling and sequential design of experiments’ methodology.

## 8. Membrane Module Modeling Checklist

Deciding which non-ideal effects to include in module performance models during the early stages of module development is a difficult task. The most comprehensive model will include all the non-ideal effects identified in the preceding section. However, the information required to perform the calculations (e.g., dependence of permeance on temperature and pressure and variability in membrane properties) may not be immediately available.

Model complexity can be reduced by initially neglecting all non-ideal effects and assuming an ideal contacting pattern based on the module geometry, e.g., assuming countercurrent contacting for a hollow-fiber module. Module performance predictions for this ideal module model provide an upper limit on performance in terms of product recovery from the feed and membrane productivity or product flow rate per unit membrane area.

To help evaluate the impact of non-idealities, a checklist was developed, as summarized in Table 4. Based on the available literature, this checklist provides guidelines for estimating the potential effects of non-idealities and how to modify the ideal model to account for them.

The potential need to use higher dimensional (i.e., 2D or 3D) models for nominal countercurrent modules can be evaluated by comparing performance predictions for countercurrent contacting with predictions for crossflow contacting. If the differences are significant over the anticipated operating range, higher dimensional models may be needed to capture more subtle effects associated with gas distribution into and from the module. Note that the observation of differences does not necessarily imply that higher dimensionality modeling is needed. It only suggests that if differences between simulation and experiment exist, they may be associated with non-ideal fluid contacting, but this does not rule out other non-ideality sources.

Module performance may depend on the variation in membrane and flow channel properties (i.e., ID, channel thickness, permeance, and selectivity) that occurs in membrane and module manufacture. Ideally, manufacturing quality controls reduce these variations to acceptable levels: <10% channel size variation and <30% permeance variation. However, if actual variability is higher, the resulting flow maldistribution effects can be detrimental to performance and should be included in module simulations.

An upper bound on the pressure drop within a module is provided by calculating pressure drops in the absence of permeation, i.e., for a constant flow. If significant, pressure drops must be calculated through the inclusion of a momentum balance, as described previously. Note that pressure drops can be minimized through changes in the module design and operating conditions selected for the desired separation [63,68].

The potential impact of Joule–Thomson effects can be determined by calculating changes in temperature associated with expansion of the feed gas to the permeate pressure, using analytical expressions or a process simulator. If significant temperature changes occur, the energy balance must be included in the simulation. Additionally, the temperature dependence of gas permeance is required. Higher temperature drops along the length of the membrane are expected at low feed temperatures and high feed pressures [23,28,33]. The extent of temperature change along the length of the membrane is also governed by the gas composition.

Concentration polarization effects may be important for gas permeances that are higher than 1000 GPU and selectivities that are larger than 100 [36]. If significant, mass transfer resistances in the contacting gas phases and the membrane porous support should be included in gas permeation rate calculations.

The potential impact of concentration- and pressure-dependent permeances is best determined from experimental permeance measurements over the anticipated range of operating conditions. Theoretical predictions of such effects may be possible using the Flory–Huggins model for rubbery polymers [31,35] or either the dual mode sorption/partial immobilization, ENSIC (ENgaged Species Induced Clustering), or NELF (Non-Equilibrium Lattice Fluid) models for glassy polymers [35,75,76,77]. These models also provide a theoretical basis for developing correlations of permeance with pressure and composition for use in module models.

The real gas behavior assumption may not be valid when operating at high pressures >10 bar [27]. An additional check on validity is provided by evaluating fugacity coefficients to determine the deviation from ideal gas behavior. If significant, a fugacity driving force is required to calculate gas permeation rates.

## 9. Conclusions

Mathematical modeling of membranes for gas separations is an important step for quantifying module performance. Simplified membrane models are likely to overpredict performance and lead to erroneous results when compared to experimental data. The development of robust mathematical membrane models for several gas separation applications that take into consideration non-ideal effects related to module manufacturing, operating conditions, and membrane properties are necessary for better process performance quantification and agreement with real data. Uncertainty quantification of membrane process models can help quantify the best estimates of uncertain design parameters through a stochastic modeling approach. However, uncertainty quantification in membrane models is still a novel process that needs more research that can lead to better model refinement strategies which can provide a better fit with the experimental data. The effective development of membrane models is essential for demonstrating the commercial competitiveness that membranes offer when comparing it to other separation alternatives through a techno-economic analysis.

A checklist for developing future simulations of modules that either possess improved membrane properties or are used in emerging separation areas is provided. As membrane permeances increase, the detrimental effect of concentration polarization within the support and in the external gas phases will have to be considered. This is the case for state-of-the-art membranes developed for CO_2_ capture and light hydrocarbon separations. Additionally, module pressure drops and internal flow distribution will be of concern when seeking module designs to minimize the energy input required to create the chemical potential driving force for permeation. These concerns are especially important in CO_2_ capture.

## Figures and Tables

**Figure 1 membranes-13-00639-f001:**
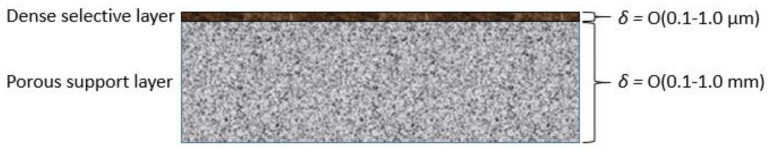
Representation of gas separation membrane, which typically consists of a thicker, porous support layer coated with a thin, dense selective layer.

**Figure 2 membranes-13-00639-f002:**
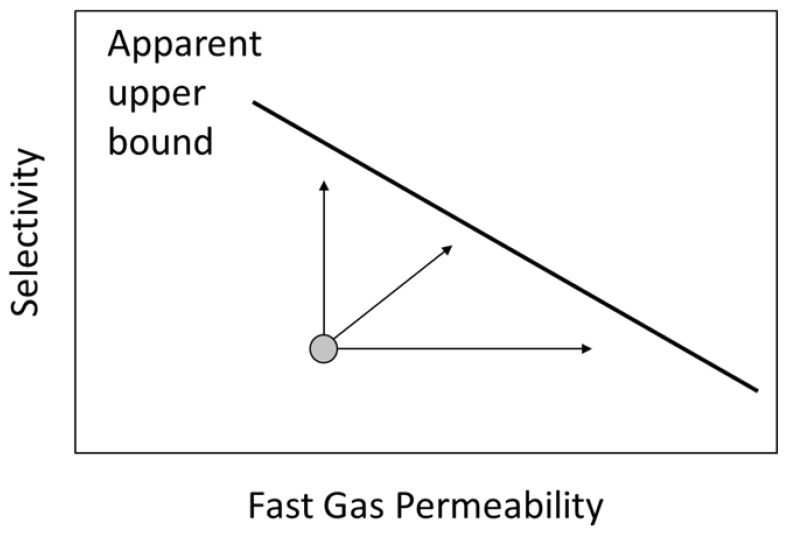
Generic upper bound observed for trade-off between selectivity (ratio of the permeability for the faster permeating species to that for the slower species) and permeability for a gas pair. This “Robeson plot” [11] implies that attempts to increase permeability, selectivity, or both (as suggested by the arrows) for a family of materials will always lie below the upper bound.

**Figure 3 membranes-13-00639-f003:**
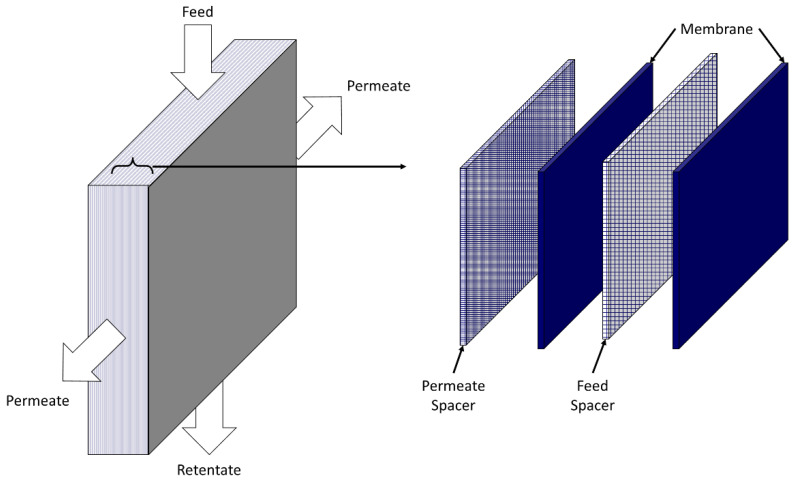
Graphic of a plate-and-frame membrane module with crossflow. Left-hand side illustrates the membrane–spacer stack and flows into and out of the stack. Right-hand side illustrates the repeating unit in the stack.

**Figure 4 membranes-13-00639-f004:**
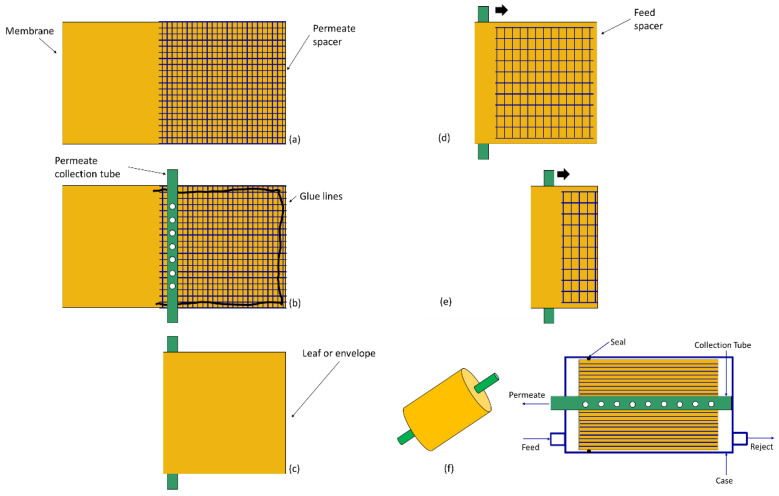
Stepwise process for construction of a spiral-wound membrane module [13]: (**a**) a permeate spacer is placed on top of the membrane; (**b**) glue is applied along the sides of the membrane, as illustrated, and the permeate collection tube is placed along the middle of the membrane sheet; (**c**) the membrane is folded over the permeate collection tube to form a leaf or envelope; (**d**,**e**) a feed spacer is placed on top of the leaf, and the permeate collection tube is rolled in the direction indicated by the black arrow to wrap the spiral-wound module; (**f**) the spiral-wound module is placed inside a case to permit fluid introduction and removal. The feed and permeate flow in a crossflow configuration with all permeate being collected by the central tube.

**Figure 5 membranes-13-00639-f005:**
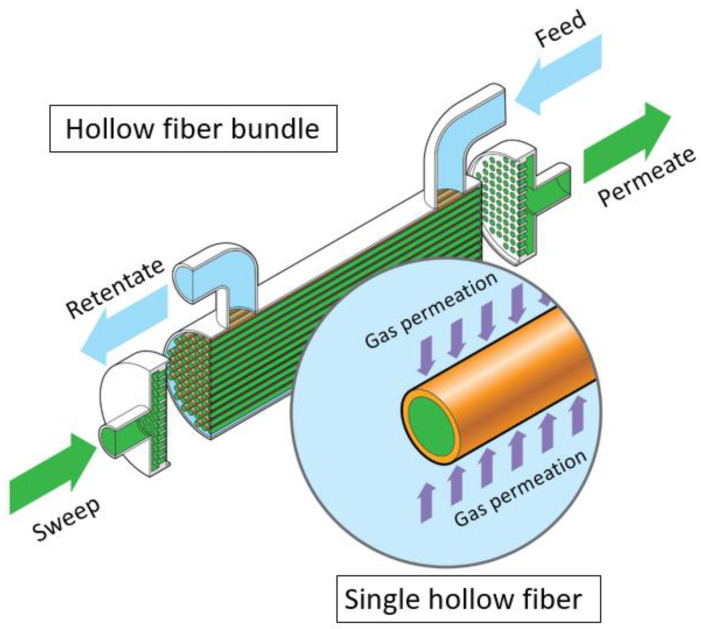
Graphic depiction of a hollow-fiber membrane module. In this configuration, the feed stream (blue) is introduced into the shell side and runs countercurrent to the sweep stream (green) introduced into the tube side (fiber lumens). The gas of interest for the separation (e.g., CO_2_ or O_2_) permeates from the feed side through the hollow-fiber membrane to the permeate side.

**Figure 7 membranes-13-00639-f007:**
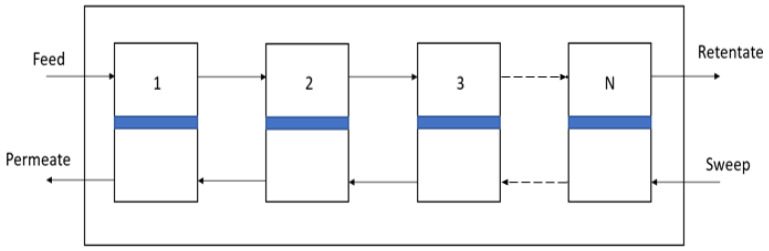
Tank in series modeling schematic for countercurrent configuration. Numbers indicate the stage number with the feed introduced in the first stage.

**Figure 8 membranes-13-00639-f008:**
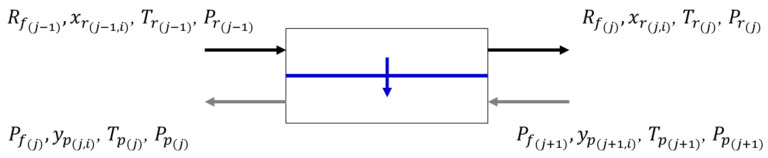
Countercurrent membrane cell. The downward arrow indicates gas permeation from the retentate to the permeate.

**Table 1 membranes-13-00639-t001:** Comparison of membrane module types (adapted from [5,12]).

	Module Type		
Metric	Hollow Fiber	Spiral Wound	Plate and Frame
Ease of Countercurrent Operation	High	Low	Low
Ease of Permeate Sweep	High	Low	Low
Pressure Drop	High (lumen side)	Moderate	Low
Concentration Polarization	High (lumen side)	Moderate	Low
Degree of Manufacturing Automation	Moderate	Low	Low

**Table 3 membranes-13-00639-t003:** Summary of several lab-scale and pilot-plant trials for multiple membrane module types. Module types listed are (1) hollow fiber (HF), (2) spiral wound (SW), and (3) flat sheet (FS).

Reference	Gas	Module Type	Pilot Scale	Process Model Validation	UQ of Model
Feng et al. [25]	O_2_/N_2_	HF	0.046 m^2^	+	-
DeJaco et al. [40]	O_2_/N_2_	SW	0.3 m^2^	+	+
Brinkmann et al. [59]	CO_2_/N_2_/O_2_	FS	12.6 m^2^	+	-
Choi et al. [60]	CO_2_/N_2_/O_2_	HF	-	+	-
Sandru et al. [61]	CO_2_/N_2_/O_2_	FS	1.5 m^2^	-	-
Hagg et al. [62]	CO_2_/N_2_/O_2_	HF	18 m^2^	-	-
He et al. [63]	CO_2_/N_2_/O_2_	HF	4.2–10 m^2^	-	-
Salim et al. [64]	CO_2_/N_2_/O_2_/H_2_O	SW	1.4 m^2^	-	-
White et al. [67]	CO_2_/N_2_/O_2_	SW	1 TPD CO_2_	-	-
Lin et al. [68]	CO_2_/CO/H_2_/N_2_/CH_4_	SW	1–40 m^2^	-	-
Scholes et al. [69]	CO_2_/N_2_/O_2_	HF and SW	5 and 7.5 m^2^	-	-
Stern et al. [70]	CO_2_/CH_4_	HF	0.93 m^2^	-	-
Pohlmann et al. [71]	CO_2_/N_2_/O_2_	FS	12.5 m^2^	-	-
Dai et al. [72]	CO_2_/N_2_/O_2_	HF	0.02 m^2^	-	-
Wolff et al. [73]	CO_2_/N_2_/O_2_	FS	11–40 m^2^	+	-

**Table 4 membranes-13-00639-t004:** Checklist for inclusion of module non-idealities.

Source of Nonideality	Check on Importance	Simulation Modification
Deviation from nominal countercurrent flow	Compare predictions for countercurrent and crossflow contacting	2D or 3D transport simulations may be required
Fiber size or channel height variation	Experimental standard deviation of size variation is >10% of average	Include fiber size or channel height variation
Membrane permeance/selectivity variation	Experimental standard deviation of variation is >30% of average	Include permeance/selectivity
Pressure drops in flow channel	Evaluate pressure drops in absence of permeation in flow channels	Include momentum balance
Joule–Thomson effects	Evaluate temperature change upon expanding feed gas to permeate pressure	Include non-isothermal permeances and energy balance
Concentration polarization	Gas permeance > 1000 GPU and selectivity > 100	Include external and internal mass transfer resistances in permeation rate calculation
Concentration/pressure dependent permeances	Experimental measurement of permeances over relevant range of pressures and compositions	Include appropriate expression of dependence of permeance on process variables
Real gas behavior	Experimental pressure > 10 bar and non-unity fugacity coefficients	Use fugacity driving force in permeation expression

## Data Availability

Not applicable.

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
