# Peer review of "Gas Separation Membrane Module Modeling: A Comprehensive Review"

_membranes, 2023, doi:10.3390/membranes13070639_

Round 1
Reviewer 1 Report
The authors made a detailed and insightful discussion on gas separation membranes, membrane modules, mathematical models and the development direction of the model.There are some problems as follows.
1. The title of this paper is "Gas Separation Membrane Module Modeling". The authors used too much content to introduce membrane and membrane module in "2. Introduction to membrane gas separations"and "3. Membrane module designs", so it is suggested to make appropriate simplification.
2. The article introduces co-current,counter-current and cross-flow, but in"5. Membrane modeling review" it seems that only the counte-rcurrent model is introduced.
3. This paper introduced the empirical equations used to correct for nonideality, and it is suggested to increase the introduction of the accuracy of these empirical equations.
Some language errors need to be corrected: Serial number problem of "1. Non-uniform membrane properties" in Line 362 ; "the value of * in Equation (1)" in Line 119; "when a pressure drop is imposed across the membrane" in Line 146 and the "pressure drop" should be "pressure difference"; and the full name of "ID" in Line 485.
Round 2
Reviewer 2 Report
The revised manuscript is compact but excellent.